# Vaccine Candidates for the Control and Prevention of the Sexually Transmitted Disease Gonorrhea

**DOI:** 10.3390/vaccines9070804

**Published:** 2021-07-20

**Authors:** Ethan C. Haese, Van C. Thai, Charlene M. Kahler

**Affiliations:** Marshall Centre for Infectious Disease Research and Training, School of Biomedical Sciences, University of Western Australia, Crawley, WA 6009, Australia; ethan.haese@research.uwa.edu.au (E.C.H.); vanchi.thai@research.uwa.edu.au (V.C.T.)

**Keywords:** *Neisseria gonorrhoeae*, gonorrhea, vaccine development, antimicrobial resistance, multi-drug resistance, sexually transmitted infections

## Abstract

The World Health Organization (WHO) has placed *N. gonorrhoeae* on the global priority list of antimicrobial resistant pathogens and is urgently seeking the development of new intervention strategies. *N. gonorrhoeae* causes 86.9 million cases globally per annum. The effects of gonococcal disease are seen predominantly in women and children and especially in the Australian Indigenous community. While economic modelling suggests that this infection alone may directly cost the USA health care system USD 11.0–20.6 billion, indirect costs associated with adverse disease and pregnancy outcomes, disease prevention, and productivity loss, mean that the overall effect of the disease is far greater still. In this review, we summate the current progress towards the development of a gonorrhea vaccine and describe the clinical trials being undertaken in Australia to assess the efficacy of the current formulation of Bexsero^®^ in controlling disease.

## 1. Introduction

Gonorrhea is a sexually transmitted infection (STI) caused by the Gram-negative bacteria *Neisseria gonorrhoeae* (also called gonococcus). There were an estimated 376 million new cases of curable STIs (chlamydia, gonorrhea, syphilis, and trichomoniasis) in 2016, 86.9 million of which were cases of gonorrhea [1]. STIs result in a substantial economic burden on individuals and society. Low-to-middle income countries often have higher estimated burdens of disease than high-income countries. Recent models indicate that sub-Saharan Africa and the Western/Eastern Pacific regions bear a disproportionate burden of 75% of global STI control costs [2]. Modelling of total costs are divided into two categories—direct medical costs for screening diagnostic tests and treatments, and lifetime costs associated with infertility which result in the need to access assisted reproductive techniques [3] and low-birth weight/preterm birth complications [4] which result in a high cost burden on public health systems. 

Mathematical models based on 2008 data from the United States of America (USA) estimated a total lifetime direct medical cost for STIs of USD 15.6 (range: USD 11.0–20.6) billion (adjusted to the USD in 2010) [5]. The estimated direct medical costs of gonorrhea infections specifically were approximately USD 162.1 (range: USD 81.1–243.2) million [5]. Another model estimated that the emergence of antimicrobial resistant (AMR) *N. gonorrhoeae* could lead to 1.2 million more gonococcal infections over 10 years in the USA alone, costing an additional USD 378.2 million [6]. However, these estimates still do not reflect the true economic burden of *N. gonorrhoeae* infections as they exclude the indirect and intangible costs associated with adverse disease and pregnancy outcomes, disease prevention, and productivity loss [3].

Effective antimicrobial treatment is essential for the prevention and control of *N. gonorrhoeae* infections, and the increased emergence of multidrug resistant (MDR) and extensively drug resistant (XDR) *N. gonorrhoeae* strains has heightened concern about the possibility of widespread untreatable gonorrhea [7]. The World Health Organization (WHO) has highlighted the urgent need for the development of new antibiotic and antivirulence treatment options and vaccines for the sustainable control of future untreatable *N. gonorrhoeae* infections [8]. The WHO and the National Institute of Allergy and Infectious Diseases (NIAID) initiated the Global Roadmap for Advancing Development of Vaccines Against STIs, which outlines the important action steps needed to advance vaccine development for STIs, including gonorrhea [9,10,11,12]. The key priority action areas from the roadmap include: (1) obtaining better epidemiological data; (2) modelling theoretical vaccine impact and cost-effectiveness; (3) advancing basic science and translational data in clinical trials; (4) defining preferred product characteristics for first-generation vaccines; and (5) characterizing the public health value of vaccines to encourage investment and guide policy decisions [11] (see [10,13] for a full review and report).

Interest in vaccine development against *N. gonorrhoeae* has been revived recently by both the increased global interest in the use of vaccines to fight AMR bacteria [9,14] and observational studies reporting that vaccines developed against the closely related pathogen *Neisseria meningitidis* (also called meningococcus) serogroup B (MenB) might provide moderate protection against gonorrhea [15,16,17]. While these studies provide promise that vaccines against *N. gonorrhoeae* are biologically feasible, they also reinforce the need to characterize the full immune response in mice and for human clinical trials to determine the efficacy of vaccine antigens.

## 2. *N. gonorrhoeae* Infection and Disease

*N. gonorrhoeae* typically colonizes the urogenital mucosa, but can also colonize extragenital mucosal sites, including the rectal or oropharyngeal mucosal epithelia (Figure 1). *N. gonorrhoeae* is easily transmitted, with a substantial proportion of individuals becoming infected after a single exposure. The estimated probability of penile-to-vaginal or vaginal-to-penile transmission is approximately 50% and 20% per sex act, respectively [18]. Estimated probabilities of transmission among gay and bisexual men or men who have sex with men (MSM) during oral and anal sex are much higher than heterosexual men, at 63% for urethral-to-pharyngeal transmission and 84% for urethral-to-rectal transmission [19].

Lower genital tract infections in men are commonly symptomatic, presenting as uncomplicated urethritis with urethral discharge of a purulent exudate and dysuria after an average incubation period of one week for heterosexual men [20,21] and four days for MSM [22]. However, for some men, clinical presentations may occur as early as 1–2 days after the last sexual contact [23,24]. Among women, genital tract infections are primarily asymptomatic or minimally symptomatic, often going unrecognized or misdiagnosed as other reproductive tract infections. When present, genital symptoms develop in most women within 10 days of exposure, manifesting as acute cervicitis with mucopurulent discharge, dysuria, vaginal pruritus, or abdominal pain [25,26]. Among both sexes, rectal and oropharyngeal infections are usually asymptomatic but can present symptomatically as proctitis and pharyngitis, respectively [27].

Without treatment more invasive forms of disease can occur, including epididymo-orchitis in men and pelvic inflammatory disease (PID) in women, or disseminated gonococcal infection with bacteremia in both sexes [28]. Most *N. gonorrhoeae* strains that disseminate do not cause urethritis, but infection of a mucosal site usually precedes cases of disseminated disease [29]. Although uncommon, disseminated disease can cause arthritis-dermatitis syndrome [30] and in rare cases endocarditis or meningitis [31,32,33,34]. Infection with *N. gonorrhoeae* can also facilitate transmission or acquisition of the human immunodeficiency virus type 1 (HIV-1) [35,36,37,38,39], and infection with *N. gonorrhoeae* has been reported to repress HIV-1 replication in macrophages inducing a state similar to viral latency [40].

PID is a common and severe complication of ascending infection of the female reproductive tract [41]. If left untreated, PID can cause scarring and dysfunction of the upper genital tract, resulting in chronic pelvic pain, ectopic pregnancy, and infertility [42,43,44]. Women with *N. gonorrhoeae* infections are also more likely to experience premature rupture of membranes, preterm birth, low birth weight, ophthalmia neonatorum (neonatal conjunctivitis), and neonatal and perinatal mortality [45]. PID has also been associated with STIs caused by *Chlamydia trachomatis*, the causative agent of chlamydia, and coinfection with *N. gonorrhoeae* and *C. trachomatis* occurs frequently [46]. However, PID caused by gonococcal infections typically presents with more severe symptoms [47].

## 3. Epidemiology of *N. gonorrhoeae* Infection and Disease in Australia

There were an estimated 86.9 million (95% uncertainty interval, UI: 58.6–123.4 million) new global incident cases of urogenital gonococcal infections in 2016, with an incidence rate of 20 per 1000 women (95% UI: 14–28) and 26 per 1000 men (95% UI: 15–41) [1]. In most countries, the rates of gonorrhea were often highest among adolescents and young adults, and higher among men than women [1,18]. In Australia, there has been a 2.3-fold increase in total gonorrhea notifications and notification rates over the last decade, up from 10,320 (46.8 per 100,000 population) in 2010 to 34,265 in 2019 (134.7 per 100,000 population) [48,49]. While increased testing is a likely factor contributing to increased notification, surveillance data suggest there is an increasing incidence of gonorrhea among heterosexual men and MSM. Notification rates in women have also increased almost 2-fold in the last decade, raising concerns about potential reproductive tract complications from asymptomatic or minimally symptomatic infections [50,51]. Additionally, the prevalence and incidence of gonococcal infections are several-fold higher among marginalized minority populations, particularly sex workers, gay, bisexual, and other men who have sex with men, trans and gender diverse people, and Aboriginal and Torres Strait Islander (Indigenous) people [18,48,50,51,52,53].

The sex workers category includes a diverse population who exchange sexual activity for income, employment, survival, or drugs, and sex workers are at increased risk of acquiring HIV-1 and STIs. While there has been a substantial increase in the incidence of oropharyngeal gonorrhea (from 1.6 to 4.9 per 100 person-year, PY) among female sex workers in Australia between 2009–2015, the incidence of urogenital gonorrhea (from 1.0 to 1.7 per 100 PY) and rectal gonorrhea (from 0.2 to 0.4 per 100 PY) has remained relatively stable [54]. After testing for oropharyngeal gonorrhea was introduced in 2017 for all female sex workers attending the Melbourne Sexual Health Centre, the prevalence of oropharyngeal gonorrhea was reported as 2.0% (95% confidence interval, CI: 1.6–2.6%) [55]. The prevalence of HIV-1 remained low and did not change over time. Among the female sex workers that tested positive for gonorrhea, 55% (95% CI: 43–67%) only tested positive in the oropharynx, highlighting the oropharynx as a reservoir for transmission and the need for testing extragenital sites of infection [55].

There are very limited data on the prevalence of gonorrhea for male sex workers compared to female sex workers. However, recent studies have estimated that the prevalence of gonorrhea among male sex workers in Australia is between 15.0% (95% CI: 12.0–19.2%) [56] and 10.8% (95% CI: 4.4–20.9%) [57]. The positivity for incident HIV-1 infections among male sex workers is between 0.6% (95% CI: 0.1–2.5%) and 1.7% (95% CI: 0.0–5.0%), with a lower positivity for HIV-1 or gonorrhea among male sex workers who exclusively have sex with women compared to those who have sex with men [56,57].

Among gay, bisexual, and other men who have sex with men the prevalence of gonorrhea is quite high. In Australia, the gonorrhea incidence among MSM increased from 14.1 per 100 PY (95% CI: 13.2–15.0%) in 2010 to 24.6 per 100 PY (95% CI: 23.9–25.4%) in 2017, with the greatest increases in oropharyngeal and rectal gonorrhea infections [58]. In contrast to the increase in the notification rate of gonorrhea, there has been a decline in the HIV-1 notification rate from 4.5 to 4.0 per 100,000 population between 2013–2017 among MSM in Australia. Most of this reduction in HIV incidence can be attributed to the introduction of pre-exposure prophylaxis (PrEP) in 2016 [59].

Trans and gender diverse is a term used to describe a group of individuals whose gender identity or presentation is different from the sex presumed for them at birth [60]. While there are limited data on STIs in the trans and gender diverse population, a health survey by the Kirby Institute reported a total rate of gonorrhea in Australian trans and gender diverse individuals of 6.4% [53]. Available data suggest a substantial prevalence of gonorrhea among transgender women, particularly at extragenital anatomical sites. The prevalence of gonorrhea among transgender women has increased from 3.1% to 9.8% over the past seven years but remained stable for transgender men [61,62].

Comparable to other countries with marginalized Indigenous populations such as the USA and Canada, the Indigenous population in Australia suffers disproportionate rates of STIs. In 2017, the rates of gonorrhea among Indigenous people (627.5 per 100,000 population) were 6.6-fold the rate for the non-Indigenous population, increasing to nearly 30-fold higher in remote and very remote communities (1442.9 per 100,000 population) [52]. The ratio of male to female notifications were almost equal compared with the number of notifications in the non-Indigenous population, which is higher among men than women [52]. Additionally, nearly three-quarters of cases of gonorrhea were notified from people aged 15–29 years among the Indigenous population compared with half in the non-Indigenous population [52]. The disparities between the two populations are most likely caused by differences in risk behaviors and testing patterns and reduced access to health services [63].

## 4. Antimicrobial Resistant *N. gonorrhoeae*

Although antibiotics have been successful in containing the prevalence of gonorrhea in the past, *N. gonorrhoeae* has successively gained resistance to all previously used first-line antibiotics to the extent that they are no longer recommended for treatment [64]. The WHO has declared *N. gonorrhoeae* a high-priority pathogen for research and development of new antibiotics because of increasing resistance to the extended-spectrum cephalosporin ceftriaxone, the last remaining option for first-line gonorrhea monotherapy [7,8,64]. In Australia, current treatment options rely on the use of ceftriaxone and the broad-spectrum macrolide azithromycin as a single dose [65].

While sporadic treatment failures of oropharyngeal gonorrhea with ceftriaxone monotherapy had been confirmed in the past [66,67,68,69,70,71,72], the first treatment failure of oropharyngeal gonorrhea with dual therapy was reported in Japan in 2016 [73]. International spread of MDR ceftriaxone-resistant gonococcal strains has since been confirmed in Japan [74], Denmark [75], the United Kingdom [76,77], France [78], Australia [79], and Canada [80]. The first XDR strain with ceftriaxone resistance and high-level azithromycin resistance was isolated in England and Australia in 2018 [81,82,83].

Existing interventions are insufficient in controlling gonorrhea and without new antibiotics or other therapeutics, there is a concern that untreatable infections will become more common in the future. To confront this issue, different approaches are being investigated, including improving antibiotic stewardship, novel drug discovery, antivirulence therapeutic approaches, and gonococcal vaccine development for the sustainable control of future *N. gonorrhoeae* infections [13,64,84,85,86,87,88].

## 5. Innate and Adaptive Immune Responses to *N. gonorrhoeae* Infection

*N. gonorrhoeae* is a highly adapted human pathogen, as individuals treated for gonorrhea can be repeatedly infected with no development of immunological memory, leading to prolonged symptomatic and asymptomatic infections [89,90,91]. Symptomatic infection is typically characterized by a purulent discharge composed of bacteria and neutrophil granulocytes. *N. gonorrhoeae* has evolved to modulate and evade host innate and adaptive immune responses (Figure 2). Experimental investigation is complicated by the fact that humans are the only known natural host. However, with the use of human and murine immortalized cell lines and an estrogen-treated murine infection model, insights into the immune modulation and evasion mechanisms utilized by *N. gonorrhoeae* have been revealed [92,93].

Phase and antigenic variation of major outer membrane surface-exposed Type IV pili and opacity (Opa) proteins [94,95,96], lipooligosaccharide epitope mimicry [97], and phagosome subversion [98,99] are important mechanisms utilized by *N. gonorrhoeae* to overcome host immune defenses. While gonococcal strains can express up to 10–12 different Opa proteins that differ among strains, Opa52 (also referred to as OpaG) from strain MS11 in particular has been reported to bind to carcinoembryonic antigen-related cellular adhesion molecule 1 (CEACAM-1, CD66a) on activated human CD4+ T lymphocytes and downregulate their activation and proliferation in response to antigens [100,101]. However, Zariri, et al. [102] showed that while Opa-CEACAM-1 binding resulted in reduced Opa-specific antibody titers in a human-CEACAM-1 transgenic mouse model, there was no difference in the immune response against outer membrane vesicles (OMVs) indicating that these antigens may not have an effect on vaccine design. *N. gonorrhoeae* also interacts with local mucosal immune cells to modulate host immune responses. Similarly to Opa, the major outer membrane porin PorB inhibits dendritic cell stimulation of CD4+ T cell proliferation [103,104].

Mucosal resident macrophages and recruited neutrophils and monocytes fail to control *N. gonorrhoeae* replication during an infection [105,106,107]. *N. gonorrhoeae* upregulates expression of immunoregulatory cytokines interleukin-1 (IL-1), IL-6, IL-8, IL-10, and tumor necrosis factor alpha (TNF-α) in human macrophages, which may stimulate inflammatory and immunosuppressive responses [108]. The T cell-stimulating factor IL-12 involved in the Type 1 helper T cell (Th1) response was only detected from challenged macrophages derived from primary human monocytes (MDM) [108]. In addition, infected murine macrophages expressed the immunoregulatory cytokines IL-10 and transforming growth factor beta (TGF-β), but not the proinflammatory cytokine TNF-α [109]. There was also no upregulation of costimulatory CD86 and major histocompatibility complex class II molecules reported, indicating that a tolerogenic phenotype was induced in the antigen-presenting cells [109]. MDM challenged with *N. gonorrhoeae* have also been reported to differentiate toward an M2 profile [110]. During an infection, macrophages are polarized into classical proinflammatory M1 or alternative anti-inflammatory/proresolving M2 activated macrophages in response to host immune mediators and infectivity or virulence factors [111]. The M1 phenotype promotes a Th1 response and possesses strong microbicidal and tumoricidal activity [112], while the M2 phenotype promotes clearance of infection, dampens inflammation, tissue remodeling, tumor progression, and possesses immunoregulatory functions [113].

During infection, *N. gonorrhoeae* selectively suppresses Th1 and Th2 responses and enhances a Th17 response, which is involved in the influx of neutrophils and the recruitment of other innate defense mechanisms (Figure 2, [114,115]). The lack of protective immunity reported following an infection with *N. gonorrhoeae* might be related to the induction of a Th17 response. Antibodies against the surface reduction modifiable protein Rmp (previously referred to as protein III) could also interfere with protective immunity as they block the activity of bactericidal antibodies targeting gonococcal surface antigens [116,117,118].

Given all these mechanisms for immune evasion and modulation, individuals can be repeatedly infected with no development of immunological memory that can prevent natural reinfection. Therefore, it is unsurprising that early vaccine attempts with a killed whole cell vaccine and single antigen pilus and PorB vaccines were unsuccessful [119,120,121,122,123,124].

## 6. Vaccine Development

Because infection induces a limited and weak immune response which is strain specific, modern approaches to vaccine development aim to develop vaccines containing broadly antigenic components with immunomodulatory adjuvants that both broaden the antibody responses to encompass a greater antigenic diversity while improving priming to infection. There are four current vaccine approaches: (1) meningococcal and gonococcal OMV vaccines that are intrinsic self-adjuvants; (2) purified protein subunit vaccines; (3) mixed OMV and protein subunit vaccines; and (4) immunotherapeutic vaccines that utilize adjuvants to stimulate Th1-specific immune responses (Table 1).

OMVs are naturally secreted vesicles formed from the invagination of the outer membrane during bacterial growth and contain many outer membrane antigens [125]. Since the antigens are embedded in their natural milieu with endotoxin, this retains structure and improves antigenicity without the need of adjuvants. For this reason, OMVs have been used as the platform for the development of many variants of the meningococcal vaccine [126]. However, these OMVs raise the best responses to homologous strains from which they are derived and so supplementation with other proteins has been used to modify them to raise strong heterologous responses. Multi-subunit vaccines containing multiple proteins or conserved proteins may also raise a broadly cross-protective immune response. Immunotherapeutic vaccines aim to stimulate or enhance the adaptive immune response to gonococcal infections using therapeutic and prophylactic strategies. IL-12 is an inflammatory cytokine that stimulates Th-1 associated immunity and enhances humoral or antibody-mediated immunity [127,128]. Preclinical studies using local administration of microencapsulated IL-12 have shown an enhancement of Th1-driven protective immunity and protection against reinfection in mice when given either as a treatment for an existing gonococcal infection [129] or as an adjuvant with a gonococcal vaccine [130,131]. Additionally, mice immunized with multi-antigenic peptide 2C7 epitope mimic combined with the toll-like receptor 4 agonist monophosphoryl lipid A shortened the time of gonococcal carriage in a female murine genital infection model through a Th1-biased immunoglobulin G subclass 2a antibody response [132,133]. The monoclonal antibody 2C7 epitope is a conserved oligosaccharide structure of LOS from *N. gonorrhoeae* [134]. While carbohydrates are poor immunogens and induce T-cell-independent immune responses, peptide mimics of carbohydrate antigens elicit broader cross-reactive immune responses and are more promising vaccine candidates [132].

Vaccine development for gonorrhea is still currently in the preclinical phase, as vaccine candidates have yet to be used in human clinical trials. Recent reviews have summarized promising antigen targets and current research and development efforts for vaccines against *N. gonorrhoeae* [10,13,84,88,93,134,135,136,137,138,139]. Promising vaccine candidates are currently being evaluated in murine infection models with different adjuvants and antigen-delivery systems. The evaluation of vaccine candidates has been challenging because no correlates of protection have been identified against *N. gonorrhoeae* in humans. Current preclinical investigations measure vaccine candidate efficacy by bactericidal or opsonophagocytic activity, blocking of target function, antibody surface binding, and coinfection with *N. gonorrhoeae* in a female murine genital tract infection model [93,137]. However, whether host responses in mice will accurately predict vaccine efficacy in humans remains unknown, and while there is a controlled human infection model it is restricted to experimental urethral infection among male volunteers [140]. This is a restriction on de novo vaccine programs since it has been well established that the initial stages of infection in women including the cell types and receptors used are entirely different from males [28].

### 6.1. Evidence of a Protective Effect from Serogroup B Meningococcal Vaccines

Optimism about the feasibility of gonococcal vaccine development has been recently revived because of accumulating observational data related to vaccines developed for preventing disease from MenB. The composition of the meningococcal polysaccharide capsule defines serogroups, with A, B, C, W, X, and Y the main cause of invasive meningococcal disease [146].

The evidence for a protective effect provided by the meningococcal VA-MENGOC-BC^®^ vaccine (Finlay Institute, Cuba) against *N. gonorrhoeae* was first reported in Cuba after the rapid decline in gonorrhea incidence following a vaccine campaign from 1988–1990 that targeted the highest-risk population (people aged 3 months to 24 years) [16,147,148]. The effectiveness of the vaccine against meningococcal disease was estimated to be between 81 and 83% after two doses, and the coverage for people under 24 years old was 95% [147]. VA-MENGOC-BC is a bivalent vaccine based on proteoliposome OMVs containing more than one hundred proteins from a hypervirulent MenB strain supplemented with meningococcal serogroup C polysaccharide [141,149]. Many conserved proteins between *N. meningitidis* and *N. gonorrhoeae* have been identified in the OMV component of the vaccine which could induce cross-reactive bactericidal antibodies against heterologous MenB strains and possibly against *N. gonorrhoeae* [16,147,150,151].

The MenB OMV vaccine MeNZB^®^ (Novartis) was introduced to the national immunization program in New Zealand in 2004. The effectiveness of MeNZB was estimated to be 77% (95% CI: 62–85%) after three doses, and the coverage for people under 20 years old was 81% [152]. MeNZB was successful in controlling the MenB epidemic and the immunization program ended in March 2011. Mathematical modeling indicated that exposure to MeNZB not only prevented meningococcal disease caused by MenB but also had an effectiveness of 31% against infection with *N. gonorrhoeae* [15]. Moreover, vaccination significantly reduced the risk of hospitalization from gonorrhea [153].

The development of the recombinant protein-based 4CMenB (Bexsero^®^, GSK) vaccine is one of the more recent advances in the prevention of invasive meningococcal disease [145]. The Bexsero^®^ vaccine contains the MeNZB OMV antigens and three additional recombinant antigens (factor H binding protein, fHbp; neisserial heparin binding antigen, NHBA; and *Neisseria* adhesin A, NadA). The vaccine was first licensed in Europe in 2013, and was later introduced to Australia, Canada, and some countries in South America for use in infants from 2 months of age [145,154]. Since 2015, Bexsero^®^ has been approved for use in the USA for people aged 10–25 years. However, it has been licensed for use against MenB outbreaks in universities in the USA since December 2013 [154,155,156]. A study in the UK found that the effectiveness of the Bexsero^®^ vaccine against meningococcal disease was 82.9% (95% CI: 24.1–95.2%) at approximately 6 months after the two-dose schedule [157]. In a retrospective study in the Saguenay–Lac-Saint-Jean region of Quebec, Canada, after a group of individuals from 2 months to 20 years of age were vaccinated in 2014, there was a 59% decline in gonorrhea notifications among people aged 14–20 years was observed during the postvaccination period, suggesting the cross-protection of Bexsero^®^ against *N. gonorrhoeae* [158]. The Bexsero^®^ vaccine induced cross-reactive human antibodies to NHBA, with a 34-fold increase between pre- and post-vaccination [159]. Immunization of estrogen-treated mice with Bexsero^®^ significantly accelerated clearance and reduced *N. gonorrhoeae* bacterial burden compared to the alum or PBS control [17]. Antibodies from immunized mice could also recognize several gonococcal outer membrane proteins, including PilQ, BamA, MtrE, PorB, and Opa [17]. These findings on the cross-protection of MenB vaccines against *N. gonorrhoeae* have supported the feasibility of vaccine development for gonorrhea.

### 6.2. Clinical Trials for Efficacy of Meningococcal Vaccines against Gonorrhea in Australia

There are two current clinical trials in Australia that are investigating the efficacy of the Bexsero^®^ vaccine in preventing gonorrhea in MSM who are at high risk of gonorrhea infection. The ‘MenGO’ study is a clinical trial sponsored by the Gold Coast Sexual Health Service in Queensland [160]. This is a Phase III randomized controlled trial of the Bexsero^®^ vaccine for the prevention of gonorrhea infection in the MSM (including cis and trans men, trans women, and nonbinary people who have sex with men), comparing the incidence of gonorrhea among vaccinated and unvaccinated participants. The ‘GoGoVax’ study is a clinical trial sponsored by the Kirby Institute at the University of New South Wales in Sydney with collaborators at Griffith University in Queensland [161]. This is a Phase III, double-blinded, randomized placebo-controlled, multicentered trial also evaluating the efficacy of the Bexsero^®^ vaccine in the prevention of gonorrhea in MSM. The primary outcomes of this study are to determine whether the Bexsero^®^ vaccine changes the incidence of the first instance of symptomatic *N. gonorrhoeae* infection of the urethra, anorectum, or vagina, and the overall incidence of all *N. gonorrhoeae* infections during the study period between vaccinated and unvaccinated participants.

Additionally, the ‘B Part of it NT’ study is an observational study sponsored by the University of Adelaide with collaborators from universities and government bodies from around Australia [162]. This study will aim to implement a targeted Bexsero^®^ immunization program in adolescents aged 14–19 years in the Northern Territory. While the primary aim of this study is to examine the carriage of *N. meningitidis* in the nasopharynx, there is a secondary observational arm which will compare the rates of gonorrhea in the vaccinated versus unvaccinated participants. This is an important study as it will directly assess the impact of Bexsero^®^ in Indigenous communities where rates of gonococcal disease are 627.5 per 100,000 population or higher, a disease burden among the highest in the developed world [52].

## 7. Conclusions

Interest in the development of a vaccine against *N. gonorrhoeae* has grown in recent years, given the increasing reports of AMR strains and the promising evidence of cross-protection of MenB vaccines indicating the biological feasibility of a gonococcal vaccine. A successful gonococcal vaccine will need to induce a protective immune response greater than that generated during natural infection and counteract or overcome the mechanisms used by the bacteria to evade the adaptive immune response. In addition, there are many implementation issues with a gonorrhea vaccine. While there has been some success for vaccines against STIs, including the hepatitis B virus and human papillomavirus, these vaccines are not promoted as STI vaccines but rather as vaccines for hepatitis/hepatic cancer and cervical cancer, respectively [163,164].

While it is promising that a population-based vaccination strategy (3 months to 24 years) of a low efficacy vaccine such as the VA-MENGOC-BC vaccine in Cuba has resulted in a reduction of gonorrhea, the tantalizing prospect of a high-efficacy vaccine delivered directly to adult at-risk groups may circumvent public perceptions but increase the cost of vaccine production.

Clearly articulating the benefits of a gonorrhea vaccine by reducing the economic burden on society in general, controlling the spread of AMR *N. gonorrhoeae* and their threat to over-the-counter treatment and improving women’s health specifically are all necessary for preparing a successful implementation strategy (see [13] for a full report).

## Figures and Tables

**Figure 1 vaccines-09-00804-f001:**
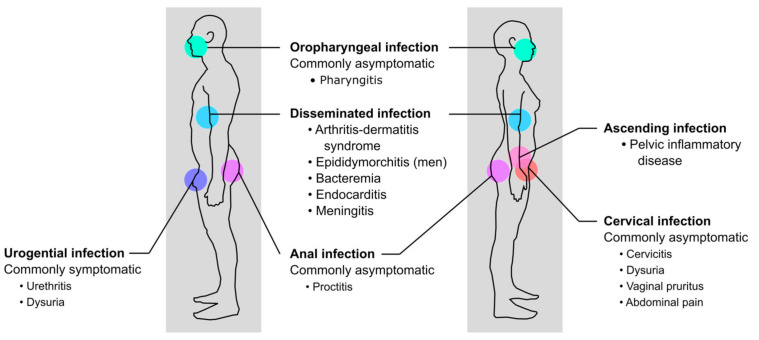
Site of infection and clinical symptoms of gonorrhea in men and women.

**Figure 2 vaccines-09-00804-f002:**
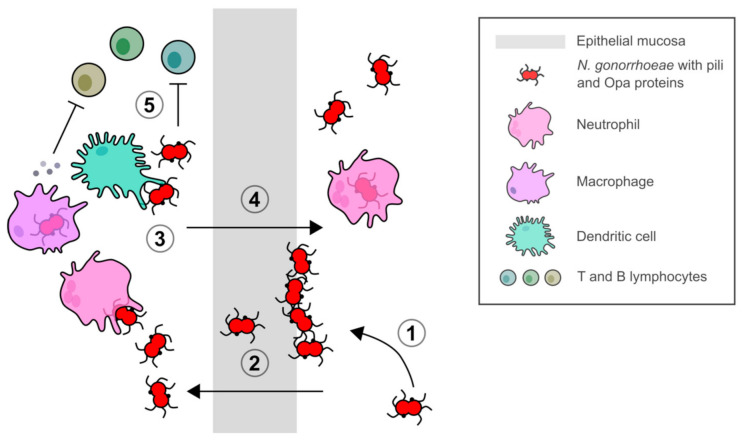
*N. gonorrhoeae* infection, transmission, and modulation of host immunity. (**1**) Attachment and colonization of *N. gonorrhoeae* mediated by Type IV pili and Opa proteins. (**2**) Colonization and invasion of host epithelium and transcytosis through epithelial mucosa. (**3**) Stimulation of local mucosal immune cells and host inflammatory response. (**4**) Transmission of bacteria in a neutrophil-rich exudate. (**5**) Modulation of immune response of T and B lymphocytes by stimulation of proinflammatory cytokines and chemokines by phagocytic cells to enhance a Th17 immune response and suppress Th1/2 immune responses.

**Table 1 vaccines-09-00804-t001:** Approaches for vaccines against *N. gonorrhoeae*.

Vaccine Approach	Vaccine Components/Antigens under Investigation ^1^	References
Meningococcal and gonococcal OMVvaccines	VA-MENGOC-BC^®^	[141]
	MeNZB^®^: NZ 98/254 OMV (Omp85, FetA, PorA, PorB3, FbpA, RmpM, OpcA, and NspA) Formalin-inactivated whole cell microparticles	[142,143]
Purified protein subunit vaccines	AniA, Lst, OmpA, Opa, OpcA, PilC, PilQ, PorB, TbpB, TbpA, TdfJ, *Ngo*Φfil phage particles	[135,144]
Mixed OMV and protein subunit vaccines	Bexsero^®^: MeNZB OMV antigens with additional fHbp, NHBA, and NadA antigens	[145]
Immunotherapeutic vaccines	OMV vaccine with IL-12 adjuvant	[130]
	2C7 LOS epitope mimic multi-antigenic peptide vaccine	[132]

^1^ Omp85, outer membrane protein assembly factor; FetA, ferric enterobactin receptor; PorA and PorB, outer membrane porin protein; FbpA, fibronectin binding protein; RmpM, reduction-modifiable protein; OpcA, outer membrane adhesin; NspA, neisserial surface protein; AniA, anaerobically induced copper-containing nitrite reductase; Lst, lipooligosaccharide-specific α-2,3-sialyltransferase; OmpA, outer membrane protein; PilC, Type IV pilus assembly protein; PilQ, Type IV pilus biogenesis and competence protein; TbpA and TbpB, transferrin-binding protein; TdfJ, outer membrane TonB-dependent transporter protein; fHbp, factor H binding protein; NHBA, neisserial heparin binding antigen; and NadA, *Neisseria* adhesin and invasin.

## Data Availability

Not applicable.

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
