# Peer review of "Vaccine Candidates for the Control and Prevention of the Sexually Transmitted Disease Gonorrhea"

_vaccines, 2021, doi:10.3390/vaccines9070804_

Round 1

Reviewer 1 Report

This work by Haese and colleagues is an interesting and useful review of the need for and development of gonorrhea vaccines. Strengths of this work include the conscientious citing of older, primary literature along with recently published data, which provides a stronger perspective on this topic. The review is also comprehensive with respect to the different topics discussed, which are directly or indirectly related to gonorrhea vaccines.

Some suggestions (mostly editorial) are:

  1. In the first paragraph, the authors present much economic data from the U.S. to illustrate the economic costs of gonorrhea, which seems odd since the title of the article is “An Australian Perspective”. The US data are interesting but not all that useful to people living in other parts of the world since medical costs and the way health care is provided in different places differs widely. Is it possible to add data from Australia or European countries, where health care costs are presumably lower and health care is more available to everyone compared to the U.S. system?

Additionally, while less documented, gonorrhea is extremely costly to low-to-middle income (LMIC) countries where infection rates are very high. Further, there are significant social costs and of not being able to treat infertility in many LMIC, and there are data that project the financial cost of a reduced future work-force in places where infertility is high due to STIs. The cost of premature delivery (as results from infection-induced premature rupture of membranes) is also out of reach for many people in these countries. Perhaps this more global perspective could be added to this introductory paragraph. Example reports on Gc or Ct-induced infertility or adverse pregnancy outcomes that might be helpful are:; Inhorn and Patrizio Human Reprod Update 2015; 21:411-426; Araoye. 2003. West Afric J Med 22:190-196; Butali et al. 2016. Pan Afr Med J 24:1; Korenromp 2017. PloS One 12:e0170773; Chessen 2017. Sex Transm Dis.

  1. Need for transition sentence for the Innate and Adaptive Immune Responses: While the topics discussed under each subheading are well researched, it would be help the manuscript to include a transition sentence to tie in the discussion on immune responses, which is indirectly related to gonorrhea vaccine development and to angle the discussion that way.

  1. Opa-mediated immunosuppression: The discussion of the reports that showed Opa52 is immunosuppressive should be explained more due to the huge problem of people using different nomenclature systems to refer to specific opa alleles over the years, and a lack of understanding that the opa gene repertoire varies widely among strains. Opa52 refers to one Opa protein in one gonococcal strain, and to my knowledge, whether the corresponding opa allele (opa52) is found in many or even any other strains has not been reported. It would be better to state that gonococcal strains express 10-12 different Opa proteins that differ among strains, and that some Opa proteins have been shown to be immunosuppressive.

Also, the authors could link this finding back to vaccine development, which would be useful to those developing gonorrhea vaccines. Specifically, Van der Ley’s group showed that the immunosuppressive effect of Opa-CEACAM1 binding resulted in reduced Opa-specific antibody titers in CEACAM1-Tg mice compared to wild-type mice, but no difference in antibody titers against gonococcal OMVs. Therefore, unless Opa proteins are a protective target, this immunosuppressive mechanism may not interfere with vaccine development (Zariri et al. doi: 10.1016/j.vaccine.2013.07.069. Epub 2013 Aug 6).

For the discussion of M1 and M2 macrophages and of inhibition of Th1/Th2 responses, how might this information should inform the choice of vaccine adjuvants that might reverse the immunosuppression (i.e.  Mike Russell’s use of microencapsulated IL-12 plus Gc OMVs).

Reviewer 2 Report

REPORT FOR TRANSMISSION TO AUTHORS

In this brief review, the authors reviewed the literature regarding the current progress towards the development of a gonorrhea vaccine. The topic is an important subject. The manuscript is nicely presented and easy to understand.

Author Response

Thank you very much for the encouraging response.

Reviewer 3 Report

COMMENTS AND SUGGESTIONS FOR AUTHORS:

This review will be of substantial interest to researchers, healthcare providers, and funding agencies. The authors have reviewed, in sufficient details, the current knowledge of the pathogen, disease, epidemiology, drug resistance, and immunopathology. The authors’ review of the ongoing vaccine development efforts is mostly limited to cross-protection by meningococcal vaccines. Nevertheless, overall, this review addresses an important topical question with implications for gonococcal disease prevention and control.

The authors should address the few major and minor comments listed below so as to meet the Journal’s requirement for publication.

MAJOR POINTS:

1) The title of this manuscript does not accurately reflect much of the contents and focus of this review article. Although the title states “Development of a Vaccine against Neisseria Gonorrhoeae: An Australian Perspective,” there is no mention of anything to this effect in the manuscript. Indeed, the authors indicate in section 5.2 (Lines 302-319) that current gonorrhea vaccine research is still in the preclinical phase of antigen discovery and identification of correlates of protection to better measure vaccine candidate efficacy. Accordingly, the focus of section 5 (Vaccine Development) is on the cross-protection provided by the Meningococcal vaccine against Gonorrhea. For instance, the two paragraphs (lines 320 - 345) under the heading “Clinical Trials for Efficacy of Meningococcal Vaccines against Gonorrhea in Australia” briefly describe the clinical trials in Australia that are assessing the efficacy of the Meningococcal vaccine Bexsero to provide cross-protection against Gonorrhea. The authors should consider an accurate title that better reflects the focus and contents of this review article.

2) Related to point #1, the authors should also consider reviewing (in a bit more detailed manner) the current status of the ongoing efforts to design and develop vaccines that provide direct protection against Neisseria Gonorrhoeae. The authors’ discussion on this pertinent topic is limited to just two sentences (lines 306-310) and to the authors directing the readers to other reviews (lines 304-306).

MINOR POINTS:

1) Figure 1

Suggested revision: The unlabeled colored circles might be confusing to some readers (especially, when the article is printed in black and white). For clarity, consider moving the “Oropharyngeal infection” and “Disseminated infection” and the list of associated symptoms to the space in between the cartoons and use connecting lines to point to the site of infection in both sexes. To accommodate these texts in the space between the cartoons, the “anal infection” and associated text could be moved a bit lower. 

2) Figure 2

Suggested revision: For clarity, consider labeling the different cell types or providing a key in the figure legend. 

3) Line 38: MDR and XDR

Suggested revision: Define the acronyms upon first time use.

4) Line 188: “4. Innate and Adaptive Immune Responses to N. gonorrhoeae infection”

Suggested revision: This section number should be 5 (not 4).

5) Line 243: “5. Vaccine Development”

Suggested revision: This section number should be 6 (not 5).

6) Line 252: “5.1”

Suggested revision: This sub-section number should be 6.1 (not 5.1).

7) Line 302: “5.2”

Suggested revision: This sub-section number should be 6.2 (not 5.2).

8) Line 320: “5.3”

Suggested revision: This sub-section number should be 6.3 (not 5.3).

9) Line 338: “This study will aim implement a targeted …”

Suggested revision: “This study will aim to implement a targeted …”

10) Line 348: “… years given increasing reports of AMR and promising …”

Suggested revision: “… years given increasing reports of AMR strains and promising …”

11) Lines 351-352: “… and counteract or overcome the mechanisms used to evade the adaptive immune response.”

Suggested revision: “… and counteract or overcome the mechanisms used by the bacteria to evade the adaptive immune response.”
